# Quantitative proteomics reveal proteins enriched in tubular endoplasmic reticulum of *Saccharomyces cerevisiae*

Xinbo Wang[1], Shanshan Li[2], Haicheng Wang[3], Wenqing Shui[2]*, Junjie Hu[4]*

[1]Department of Genetics and Cell Biology, College of Life Sciences, Nankai University, Tianjin, China; [2]iHuman Institute and School of Life Science and Technology, ShanghaiTech University, Shanghai, China; [3]College of Life Sciences, Huazhong University of Science and Technology, Wuhan, China; [4]National Laboratory of Biomacromolecules and CAS Center for Excellence in Biomacromolecules, Institue of Biophysics, Chinese Academy of Sciences, Beijing, China

**Abstract** The tubular network is a critical part of the endoplasmic reticulum (ER). The network is shaped by the reticulons and REEPs/Yop1p that generate tubules by inducing high membrane curvature, and the dynamin-like GTPases atlastin and Sey1p/RHD3 that connect tubules via membrane fusion. However, the specific functions of this ER domain are not clear. Here, we isolated tubule-based microsomes from *Saccharomyces cerevisiae* via classical cell fractionation and detergent-free immunoprecipitation of Flag-tagged Yop1p, which specifically localizes to ER tubules. In quantitative comparisons of tubule-derived and total microsomes, we identified a total of 79 proteins that were enriched in the ER tubules, including known proteins that organize the tubular ER network. Functional categorization of the list of proteins revealed that the tubular ER network may be involved in membrane trafficking, lipid metabolism, organelle contact, and stress sensing. We propose that affinity isolation coupled with quantitative proteomics is a useful tool for investigating ER functions.

*For correspondence: shuiwq@ shanghaitech.edu.cn (WS); huj@ ibp.ac.cn (JH)

**Competing interests:** The authors declare that no competing interests exist.

## Introduction

The tubular endoplasmic reticulum (ER) network is a morphologically unique part of the ER that consists of interconnected cylindrical membrane structures, with a diameter of ~30 nm in yeast and ~50 nm in higher eukaryotes (*Hu et al., 2011*). In this network, tubules constantly form, retract, move, and fuse with one another (*Du et al., 2004*; *Griffing, 2010*). The abundance of the tubular network varies between different cell types. In *Saccharomyces cerevisiae*, electron microscope (EM) tomography has revealed that ~30% of the ER is composed of the tubular network (*West et al., 2011*). In mammalian cells, the tubular ER network is more prominent in cell lines, including COS-7 and U2OS cells. The distribution of the network also varies. In yeast and plant cells, the tubular ER is mostly found beneath the plasma membrane, a region termed the cortical ER (*Griffing, 2010*; *Prinz et al., 2000*). In mammalian cells, ER tubules are frequently found in the cell periphery, but there is usually a cluster of tubules in the perinuclear region (*Baumann and Walz, 2001*; *Shibata et al., 2006*). The abundance and distribution of the tubular ER network is tightly regulated in the cell.

ER tubules have high membrane curvature on cross section. Such curvature is generated and stabilized by a class of integral membrane proteins, the reticulons (RTNs) and DP1/Yop1p (*Hu et al., 2008*; *Voeltz et al., 2006*). Deletion of these proteins causes a loss of tubules and a corresponding gain in sheets, and purified reconstituted Yop1p and Rtn1p generate tubules in vitro (*Hu et al.,

*2008*). The most conserved region of these proteins is the reticulon homology domain (RHD) (*Shibata et al., 2008*; *Voeltz et al., 2006*), which contains two tandem transmembrane (TM) segments. Each of these segments consists of 30–35 amino acids, which is too long to traverse the membrane once but too short to traverse it twice; therefore, they likely form transmembrane hairpins (TMHs). The hairpin-like configuration occupies more space in the outer leaflet than the inner leaflet and may bend the membrane. In addition, these tubule-forming proteins form homo- or hetero-oligomers (*Shibata et al., 2008*), which could serve as curved scaffolds. ER tubules can also be pulled out from a flat membrane by either attaching to the growing tip of the microtubule or interacting with motor proteins (*Shibata et al., 2009*; *Zimmerberg and Kozlov, 2006*). In these cases, the tip of a newly formed tubule is often marked by Rab10 in complex with phosphatidylinositol synthase (PIS) and CEPT1 (*English and Voeltz, 2013*).

Another feature of ER tubules is that they connect to a reticular network. Fusion between tubules is mediated by a class of dynamin-like GTPase, atlastin (ATL) (*Hu and Rapoport, 2016*). Depletion of ATL results in an unbranched ER phenotype (*Hu et al., 2009*; *Rismanchi et al., 2008*), and purified and reconstituted *Drosophila* ATL fuses vesicles in vitro (*Bian et al., 2011*; *Orso et al., 2009*). No ATL is found in yeast, but a functional ortholog, Sey1p, has been identified. ATL and Sey1p are exchangeable in the maintenance of ER morphology in cells (*Anwar et al., 2012*). Structural and biochemical analysis has revealed that GTP binding induces dimerization, and hydrolysis-driven conformational changes are important for the fusion by ATL and Sey1p (*Bian et al., 2011*; *Byrnes et al., 2013*; *Byrnes and Sondermann, 2011*; *Yan et al., 2015*). Both GTPases contain a TMH near the C-terminus that interacts with the RHD of tubule-forming proteins and facilitates the localization of these ER fusogens in the tubular ER (*Hu et al., 2009*).

Two Rtns and one Yop1p have been identified in yeast, and four Rtns and six REEPs (DP1 being REEP5) (*Shibata et al., 2008*), each with variable isoforms, have been identified in mammals. Sey1p is the only well-characterized ER fusogen in yeast, and three ATLs have been identified in mammals (*Rismanchi et al., 2008*; *Zhu et al., 2003*). Deletion of these key ER-shaping proteins has revealed the importance of the tubular ER network. Loss of Rtns and Yop1p in *S. cerevisiae* results in retarded growth (*Voeltz et al., 2006*), and in *Caenorhabditis elegans* it significantly decreases the embryo survival rate (*Audhya et al., 2007*). Deletion of Sey1p in *Candida albicans* results in decreased virulence (*Yamada-Okabe and Yamada-Okabe, 2002*). Deletion or mutation of RHD3, a homolog of Sey1p in *Arabidopsis thaliana*, yields short root hairs and severe growth defects (*Zhang et al., 2013*). Furthermore, depletion of ATL in *Drosophila melanogaster* and *Danio rerio* causes neuronal defects (*Fassier et al., 2010*; *Lee et al., 2009*). In humans, mutations in ATL1 cause hereditary spastic paraplegia (HSP), a neurodegenerative disease (*Salinas et al., 2008*). However, the specific functions of the tubular ER network are not known and the role of the network in these defects is unclear.

Proteomic analysis of organelles provides significant insights into their functions. Total ER fractions, in the form of microsomes, have been studied extensively (*Gilchrist et al., 2006*; *Kanaeva et al., 2005*; *Klug et al., 2014*). In addition, rough ER (i.e., with surface ribosomes) and smooth ER have been isolated and subjected to proteomic profiling (*Gilchrist et al., 2006*). A total of 832 proteins have been identified as ER proteins in rat liver samples (*Gilchrist et al., 2006*) and 294 proteins as microsomal proteins in *Pichia pastoris* (*Klug et al., 2014*). Though proteins important for sheet formation have been revealed by comparative analysis of rough microsomes (*Shibata et al., 2010*), the proteome of ER tubules has yet to be reported. Here, we isolated tubular ER via immunoprecipitation against a tubule-specific marker. Quantitative proteomic analysis by isotope-labeling mass spectrometry revealed 79 proteins that are enriched in tubular ER. These findings suggest specific roles of the tubular ER network and provide important tools for further functional studies of ER tubules.

## Results

### Immunoisolation of ER tubule-derived microsomes

To isolate microsomes that originate from ER tubules, we designed an immunoisolation protocol using tagged-Yop1p as a specific grip. CEN vectors expressing Yop1p-2xFlag and Sec63p-HA under the control of corresponding endogenous promoters were transformed into *S. cerevisiae* (*Figure 1—*

figure supplement 1A). Yop1p marks the tubular ER network (Voeltz et al., 2006) and Sec63p, a component of the translocon-associated complex, marks the entire ER. To test whether the presence of these ectopically expressed proteins compromises ER health, we monitored the unfolded protein response (UPR) in transformed cells. No obvious HAC1 (an UPR-regulated transcription activator) processing or Kar2p (an ER luminal chaperone) upregulation, commonly used indicators of the UPR in yeast (Cox and Walter, 1996), was observed (Figure 1—figure supplement 1B,C). These results suggest that the ER in cells expressing Yop1p and Sec63p is suitable for further analysis.

ER components, including membranes, membrane-associated proteins, and luminal substances, were collected by standard cell fractionation. Briefly, yeast cells were treated with snailase to remove cell walls, broken by homogenization, and subjected to step centrifugation (Figure 1A). The nuclei and unbroken cells were eliminated by a low speed spin (1000 x g). Heavy membranes, including mitochondria, were then separated by a median speed spin (20,000 x g). Finally, microsome-enriched fractions were obtained as a pellet (P100K) in a high speed spin (100,000 x g) with the cytosol in the supernatant (S100K).

The efficiency of cell fractionation was confirmed by immunoblotting using antibodies against makers of various subcellular compartments (Figure 1B). Pma1p, a proton pump localized in the plasma membrane, and porin, a mitochondrial β-barrel protein, were largely diminished from S20K. A small amount of Pma1p reappeared in P100K, likely due to the tight association between the plasma membrane and cortical ER. Surprisingly, Golgi-targeted Tlg2p and vacuole-bound ALP remained in S100K. It is possible that the two organelles were ruptured into small vesicles under the conditions used here. Nevertheless, these markers and cytosolic kinase PGK1 were clearly excluded from P100K. Some of the endosomal marker, Pep12p, precipitated in P100K, possibly due to an association with ER membranes or its size, which was similar to that of microsomal vesicles.

ER-resident proteins, including Sec63p (integral membrane protein), Dpm1p (tail-anchored protein), Kar2p (soluble luminal protein), and Sey1p and Yop1p (known ER tubule proteins), were found mostly in P100K. As expected, some Kar2p leaked into the cytosol due to ER breakage prior to microsome formation. Approximately half of the Kar2p protein appeared in P100K, likely captured by sealing microsomal membranes. Notably, a small portion of Flag-tagged Yop1p was found in S100K. Some Yop1p may exist in ultra-light microsomes or fail to be integrated into ER membranes. Similar phenomena were observed previously when Yop1p was expressed and purified from yeast cells (Hu et al., 2008). These Yop1p proteins and associated components were excluded from further analysis. Resuspended P100K was designated as the total ER fraction (TO).

To test whether Yop1p-containing microsomes represent a subset of the ER, we performed a Percoll density gradient using TO samples (Figure 2A, upper panels, quantified in Figure 2B; a repeat shown in Figure 2—figure supplement 1A,B). Yop1p-2xFlag was observed in some, but not all, ER fractions (labeled by HA-tagged Sec63p) and peaked in fractions 3–9, indicating an association with low-density materials. In contrast, Sec63p co-existed in Yop1p-containing samples but was enriched in later fractions (12–15) with higher densities. When total proteins from the gradient were visualized by silver staining (Figure 2—figure supplement 1C), some proteins exhibited similar patterns to those of Yop1p (band #1) and Sec63p (band #3); others (bands #2 and 4) exhibited an additional pattern of distribution. These results suggest that Yop1p-containing microsomes correspond to only part of the microsomal population, probably those derived from ER tubules, and are generally lighter than microsomes derived from ER sheets.

To isolate Yop1p-containing microsomes, we incubated total microsomes with anti-Flag antibody-conjugated agarose gel. Detergents were avoided during precipitation so that the entire composition of the Yop1p-containing microsomes, including neighboring membrane proteins and luminal proteins, could be captured. The precipitation was less complete in the absence of commonly used IP detergents (like Triton X-100), as only two-thirds the total Yop1p-2xFlag was observed in the precipitates (Figure 1C). Some of the ER proteins, such as Kar2p and Sey1p, co-precipitated. However, proteins known to be enriched in ER sheets, represented by Sec63p, were detected in the precipitates, though at a much lower percentage of total protein. In addition, the Pep12p and Pma1p that remained in P100K were absent from the precipitates (Figure 1D). These results support the successful isolation of tubular ER from the total ER and other contaminations. Therefore, the anti-Flag affinity gel-bound samples were designated the tubular ER fractions (TUs).

To release microsomal components from the Flag affinity gel for subsequent proteomic analysis, we either added a large amount of the Flag peptide to elute the microsomes as a whole (Figure 1—

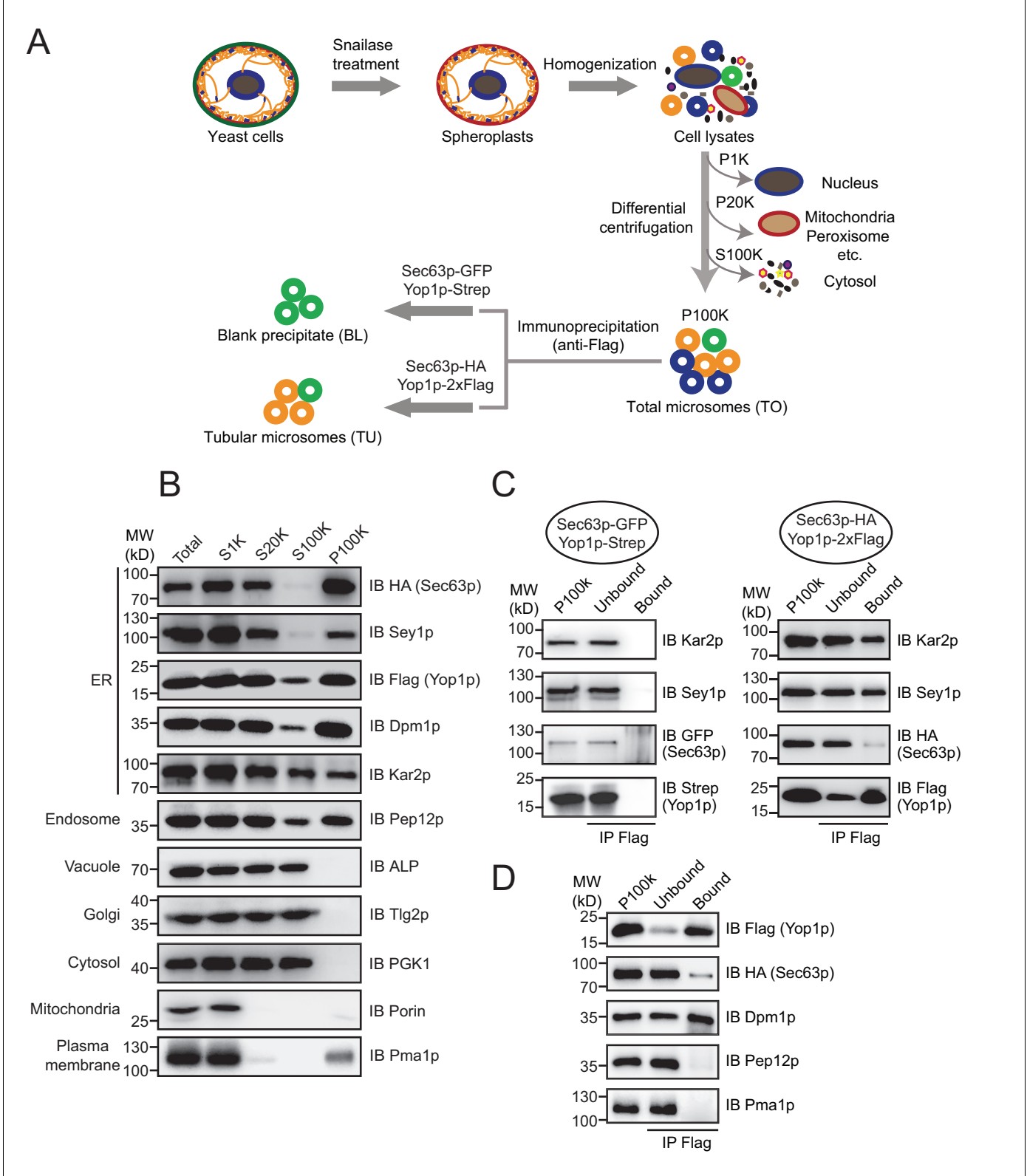

**Figure 1.** Immunoisolation of Yop1p-containing microsomes. (**A**) Schematic diagram of tubular ER isolation. P1K, P20K, and P100K represent the pellets of solutions centrifuged at 1000 x *g*, 20,000 x *g*, and 100,000 x *g*, respectively. S100K is the supernatant of the solution centrifuged at 100,000 x *g*. (**B**) Microsome preparation by step centrifugation. Samples from each step were immunoblotted using the indicated antibodies. Total, total cell lysates; S1K, S20K, S100K, and P100K are defined as in (**A**). (**C**) Sec63p-GFP/Yop1p-Strep or Sec63p-HA/Yop1p-2xFlag was co-transformed into yeast cells.

*Figure 1 continued on next page*

*Figure 1 continued*

Microsomes were prepared and subsequent immunoprecipitation (IP) performed using anti-Flag agarose beads. The samples were analyzed by SDS/PAGE and immunoblotting (IB). (D) As in (C), but with the addition of other organelle makers. The data shown in B-D are representative of at least three repetitions.

The following figure supplement is available for figure 1:

**Figure supplement 1.** Protein expression and immunoprecipitated sample elution.

*figure supplement 1D*), or simply dissolved the precipitates with a mass spectrometry-compatible detergent, RapiGest (*Figure 1—figure supplement 1E*). Elution and dissolution are more efficient with RapiGest than the peptide, yielding more than 90% of the bound proteins. Peptide-eluted microsomes were subjected to a Percoll density gradient (*Figure 2A*, lower panels). As expected, Yop1p signals appeared in the same fractions (3–9) as in experiments when total microsomes were tested. The residual Sec63p-HA signals co-migrated with Yop1p and were devoid of the high-density

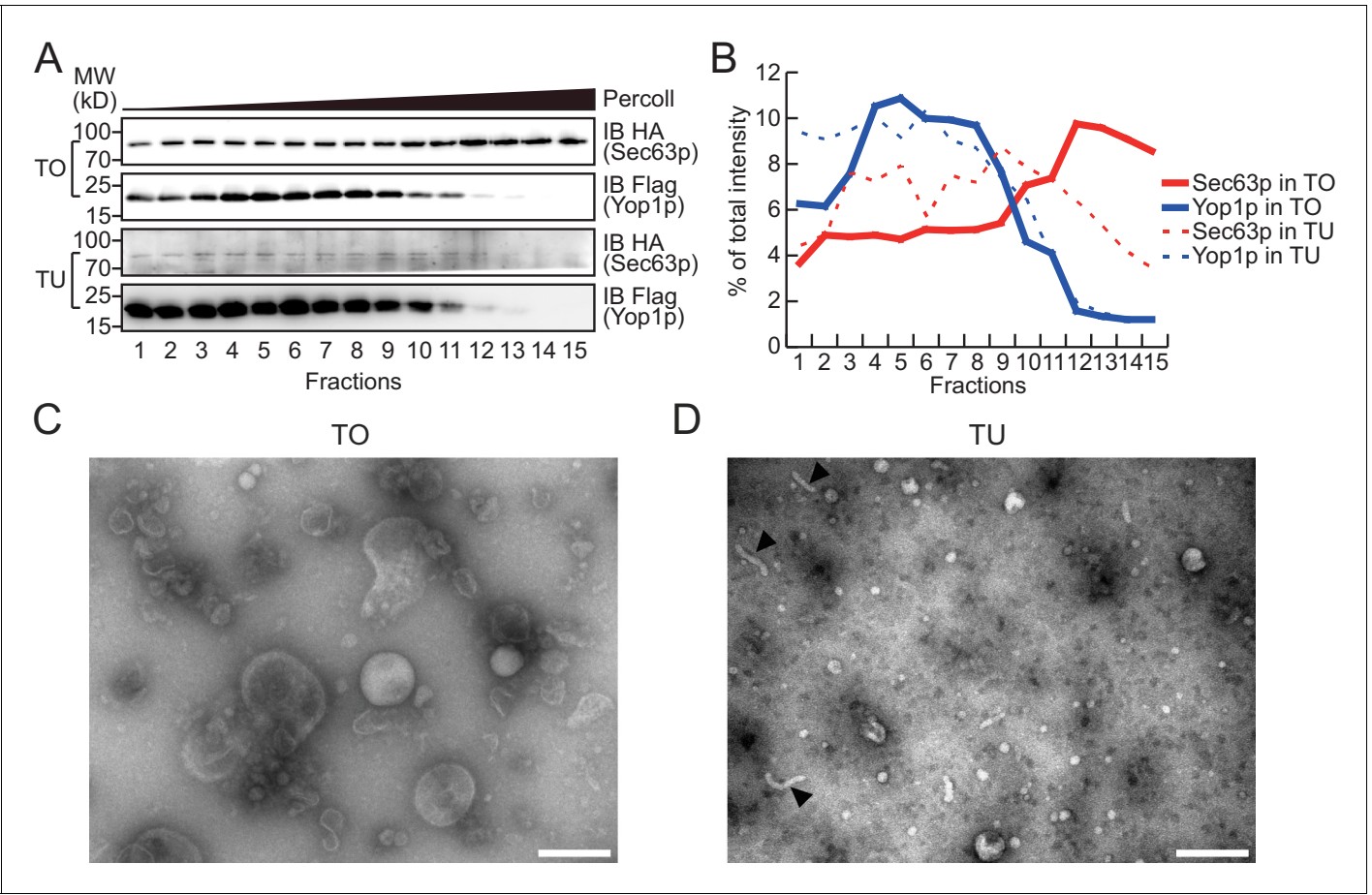

**Figure 2.** Density gradient of isolated microsomes. (A) Total microsomes (TO) or tubular microsomes (TU) were layered on 30% Percoll solution. After centrifugation for 40 min, the fractions were analyzed by SDS/PAGE and immunoblotting performed with HA or Flag antibodies. (B) Quantification of the Western blot data from (A). The data are representative of three repetitions. (C) A TO sample was analyzed by negative-stain EM. Scale bar = 200 nm. (D) A TU sample was analyzed by negative-stain EM. Arrowheads indicate tubule-like structures. Scale bar = 200 nm.

The following figure supplement is available for figure 2:

**Figure supplement 1.** Analysis of TO and TU samples by density gradients and EM.

region (12–15). We further analyzed TO and TU microsomes using negative-stain EM. TO samples exhibited microsomes of various sizes (*Figure 2C*) with diameters ranging from 200 nm to 20 nm, whereas peptide-eluted TU samples contained mostly small microsomes with occasional short tubules (*Figure 2D*). The diameters of structures seen in TU were ~20–30 nm, which is consistent with the reported size of the tubular ER in yeast cells (*Bernales et al., 2006*). These results validated the isolation of TUs.

## Proteomic analysis of ER tubule-enriched protein constituents

To identify ER tubule-enriched proteins with higher specificity, proteins/microsomes that non-specifically attach to the anti-Flag affinity gel need to be excluded. To this end, we performed the same immunoisolation using the TO from cells expressing Strep-tagged Yop1p and GFP-tagged Sec63p, and the protein precipitates from the RapiGest elution were referred to as blank precipitates (BLs). Proteins significantly more abundant in TUs than BLs were considered specific hits. Very little Yop1p-Strep, Sec63p-GFP, Kar2p, or Sey1p was observed in the BLs (negative control) (*Figure 1C*).

In the presence of equal amounts of TO, TU, and BL, proteins that exhibit a positive abundance ratio of TU:TO (>1) are likely enriched in tubular ER. Similarly, proteins with a large ratio of TU:BL (>1) are likely specifically precipitated by anti-Flag beads and related to tubular microsomes. Thus, proteins with both ratios above the thresholds can be considered putative constituents of ER tubules. We performed a quantitative proteomic analysis using the dimethyl isotope labeling technique to compare the proteome profiles between these three types of samples (*Figure 3A*). The relative TU:TO and TU:BL ratios were determined for proteins from biological triplicates (*Figure 3— source data 1*). A total of 845 proteins were identified with a quantifiable TU:TO ratio, and 531 were identified with a TU:BL ratio (*Figure 3B*). Reproducibility was demonstrated by the median CV % of protein ratio measurements across biological triplicates (17.5% for proteins with TU:TO ratios and 18.5% for proteins with TU:BL ratios). Furthermore, >90% of the proteins with TU:TO ratios (*Figure 3—source data 2-1*) or TU:BL ratios (*Figure 3—source data 2–2*) had a CV% < 30%, indicating adequate reproducibility of our proteomic quantification. Full list of protein identification and quantification with statistics is summarized in source data 1 and 2. A total of 466 proteins were quantified in both TU:TO and TU:BL comparative experiments (*Figure 3B*). To isolate candidates for the tubular ER, we retained 231 proteins with an average TU:TO ratio >2.0 (p<0.05). The list was then narrowed down to 48 proteins with an average TU:TO ratio >2 (p<0.05) and TU:BL ratio >2.0 (p<0.05), which are assumed to be specifically enriched in ER tubules (*Figure 3C,D* and *Figure 3—source data 3*). In addition, 31 proteins with an average TU:TO ratio >2.0 (p<0.05) and TU:BL ratio >1.4 (p<0.1) are candidates for likely being enriched in ER tubules (*Figure 3—source data 3*). Less stringent criteria (p<0.1 instead of 0.05) were employed for candidates with 2.0 > TU:BL ratio >1.4, considering the inclusion of proteins that may weakly associate with ER tubules. Most proteins with an average TU: BL ratio <1.4 were highly abundant cytosolic enzymes or ribosomal proteins and discarded due to their non-specific association with the Flag affinity gel beads. Notably, for the remaining 66 proteins with a TU:TO ratio >2.0 (p<0.05) and not yet quantifiable in the TU:BL comparison (*Figure 3D* and *Figure 3—source data 4*), whether they are tubular ER-specific or the result of IP contamination warrants further investigation.

## Classification and interaction network of ER tubule-enriched protein constituents

Among the 79 candidates, several proteins, such as Yop1p, Rtn1p, Rtn2p, and Sey1p, are known to have important roles in forming the tubular ER network. In addition, we identified an array of new proteins that are possibly enriched in ER tubules and may contribute to their formation and function (*Figure 3D*).

Protein localization was investigated by both a database search and manual curation (noted in *Figure 3—source data 3*). Half of the proteins from this inventory are known to localize to the ER, confirming the specificity of the isolation. The rest of the proteins are distributed in the mitochondria, nucleus, Golgi apparatus, and cytoplasm, likely reflecting the complex contacts between ER tubules and other cellular compartments.

Functional classification of this protein inventory revealed that popular categories include vesicular transport and ER organization, organelle contact, lipid metabolism, signaling/stress sensing, and

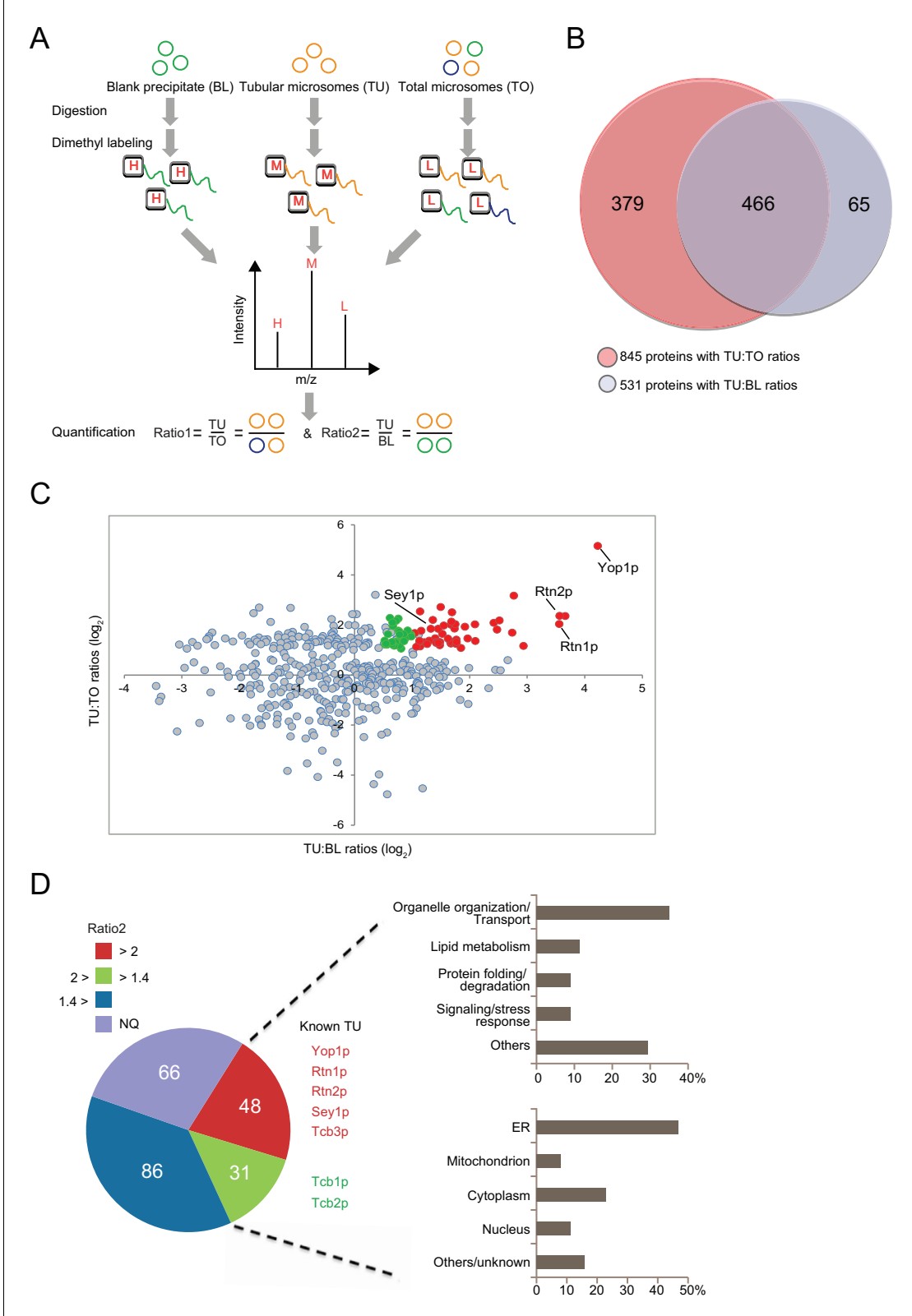

**Figure 3.** Proteomic analysis of ER tubule-enriched proteins. (**A**) Workflow of the quantitative proteomic analysis using the dimethyl isotope labeling technique. H, M, and L represent different isotope labels. (**B**) Overlap of proteins with TU:TO ratios (red) and TU:BL ratios (blue). (**C**) Relative ratio distribution of 466 proteins shown in (**B**). Proteins with TU:TO > 2.0 (p<0.05) and TU:BL > 2.0 (p<0.05) are shown in red, proteins with TU:TO > 2.0 (p<0.05) and 2.0>TU:BL >1.4 (p<0.1) are in green, and the rest are in grey. Several known ER tubule proteins are labeled. (**D**) Classification of putative

*Figure 3 continued on next page*

*Figure 3 continued*

tubular ER proteins. The pie chart includes all proteins with TU:TO > 2.0 (p<0.05) according to the range and p-values of their TU:BL ratios. The red fraction includes proteins with TU:BL > 2.0 (p<0.05), the green fraction includes proteins with 1.4< TU:BL < 2.0 (p<0.1), the blue fraction includes proteins with TU:BL < 1.4 or p-values over the thresholds for the previous two fractions. NQ, not quantifiable from the isotope-labeling MS data. For 79 putative tubular ER proteins (in the red and green regions), categorization of their major biological processes and cellular localization is shown in the upper and lower right, respectively.

The following source data and figure supplement are available for figure 3:

**Source data 1.** Protein and peptide identification.
**Source data 2.** Data summary for TU:TO and TU:BL measurements.
**Source data 3.** GO classification of putative tubular ER proteins.
**Source data 4.** List of proteins with TU:TO > 2 (p<0.05) but TU:BL not quantifiable (NQ).
**Figure supplement 1.** Protein-protein interaction map for the putative tubular ER components using the STRING database.

protein folding/degradation (*Figure 3D* and *Figure 3—source data 3*). Protein-protein interactions were also analyzed (*Figure 3—figure supplement 1*). Consistent with previously reported physical and genetic interactions, components of several protein complexes, including 'Yop1p-Sey1p-Rtn1p-Pom33p'/ 'Yet1p-Yet3p-Scs2p-Opi1p'/ 'Erv25p-Erp1p-Erp2p-Emp24', were co-identified in our experiments.

## Tests of tubular ER-enriched candidates

Next, we verified the localization of candidates identified by our proteomic studies. For clarity, we chose COS-7 cells, which have a prominent tubular ER network. Candidates with relatively confident ratios and identifiable mammalian homologs were selected first (*Table 1*). Among the six proteins, four are predicted to be integral membrane proteins (HT008, NSDH, FDFT1, and VAPA) and two cytosolic proteins (DPM1 and EMC2) (*Christianson et al., 2011*; *Colussi et al., 1997*); known functions include lipid metabolism, organelle contact, and protein processing (*Table 1*).

HA-tagged candidates were transfected into COS-7 cells, stained by anti-HA antibodies, and their localization compared to calreticulin, a total ER marker, or climp63, an ER sheet marker. In untransfected cells, the ER was present throughout the cells, with sheets at the perinuclear region and a tubular network at the cell periphery (*Figure 4—figure supplement 1A*). Climp63 signals were mostly covered by those of calreticulin. When HT008 (homolog of Nvj2p/YPR091C) and NSDHL (homolog of Erg26p) were over-expressed, they mostly aligned with calreticulin (*Figure 4A,C*) but

**Table 1.** Candidates selected for verification in mammalian cells.

| Yeast | Human | Description* |
|---|---|---|
| NVJ2 (YPR091C) | HT008 | Lipid-binding ER protein, enriched at nucleus-vacuolar junctions; may be involved in sterol metabolism or signaling. |
| ERG26 | NSDHL | ERGosterol biosynthesis, catalyzes the second of three steps required to remove two C-4 methyl groups from an intermediate in ergosterol biosynthesis. |
| ERG9 | FDFT1 | ERGosterol biosynthesis, squalene synthase; joins two farnesyl pyrophosphate moieties to form squalene in the sterol biosynthesis pathway. |
| SCS2 | VAPA | Integral ER membrane protein, regulates phospholipid metabolism; one of 6 proteins (Ist2p, Scs2p, Scs22p, Tcb1p, Tcb2p, Tcb3p) that connect ER to plasma membrane (PM) and regulate PI4P levels. |
| DPM1 | DPM1 | Dolichol phosphate mannose synthase of ER membrane; required for biosynthesis of glycosyl phosphatidylinositol (GPI) membrane anchor, as well as O-mannosylation and protein N- and O-linked glycosylation. |
| EMC2 | EMC2 | Member of conserved ER transmembrane complex; required for efficient folding of proteins in the ER. |

*From the Saccharomyces Genome Database (SGD).

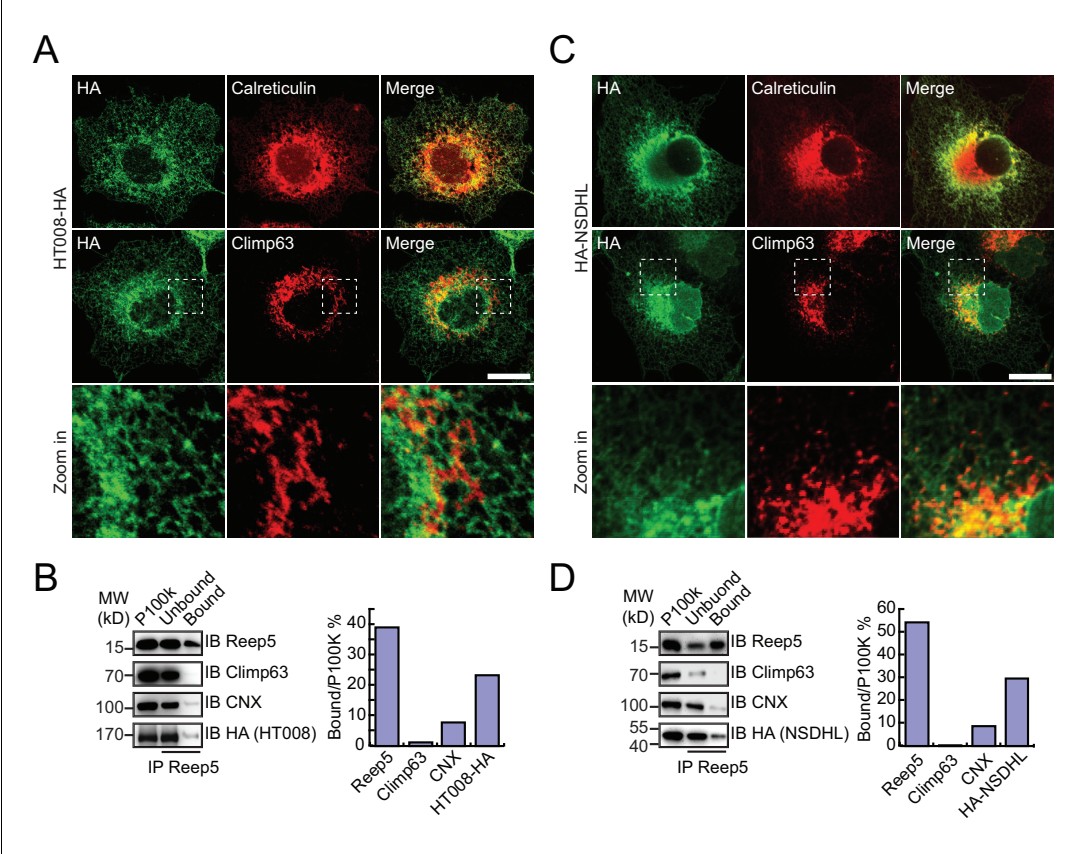

**Figure 4.** Verification of tubular ER candidates in mammalian cells. (**A**) HA-tagged HT008 was transfected into COS-7 cells. Localization was investigated using anti-HA antibodies (green) and compared to total ER protein (calreticulin, red) or ER protein sheets (climp63, red) by indirect immunofluorescence and confocal microscopy. Bottom: enlargements of the boxed regions. Scale bars = 20 μm. (**B**) HA-tagged HT008 was transfected into HeLa cells. Microsomes were isolated and immunoprecipitation performed with anti-REEP5 antibodies. The samples were analyzed by SDS/PAGE and immunoblotting (left). Quantitative results (right) were obtained based on the Western blot results and analyzed by Image J. The data are representative of at least three repetitions. CNX, calnexin. (**C**) As in (**A**), but with HA-NSDHL. (**D**) As in (**B**), but with HA-NSDHL.

The following figure supplements are available for figure 4:

**Figure supplement 1.** Verification of tubular ER candidates.

**Figure supplement 2.** Cytosolic proteins associated with the ER.

**Figure supplement 3.** Verification of endogenous tubular ER candidates using Flp-In cell lines.

**Figure supplement 4.** Verification of uncharacterized proteins in mammalian cells.

were largely missing from Climp63-positive regions (*Figure 4A,C*). These results suggest that HT008 and NSDHL are enriched in ER tubules. A similar distribution was observed when VAPA (Scs2p) or FDFT1 (Erg9p) was tested (*Figure 4—figure supplement 1B,C*). As expected, DPM1 and EMC2 were diffused in the cytosol but exhibited some ER association (*Figure 4—figure supplement 2B, C*).

To confirm the association of candidate proteins with the ER, microsomes were isolated from cells expressing these proteins via stepped centrifugation. All tested proteins were found in the P100k fraction (*Figure 4—figure supplement 2A,B*), suggesting that they co-purified with the ER. Immuno-precipitation of microsomes was then performed using antibodies against REEP5 (also known as DP1, homolog of Yop1p) and the percentage of precipitation analyzed. As predicted and similar to REEP5, the tested proteins were enriched in the precipitates. In contrast, Climp63 was barely

detected in the precipitates, and calnexin, an ER-bound chaperone, only partially co-precipitated with tubular microsomes (*Figure 4B,D* and *Figure 4—figure supplement 1D,E*). To further confirm their distribution at endogenous levels, we generated a Flp-In-293 cell line expressing REEP5-Flag (*Figure 4—figure supplement 3A,B*) and immunoisolated TU microsomes using anti-Flag affinity gel. Endogenous HT008, NSDHL, and VAPA were found in P100K (*Figure 4—figure supplement 3C*) and enriched in REEP5-Flag-labeled precipitates (*Figure 4—figure supplement 3D*). These results confirmed that proteins identified by our methods are enriched in the tubular ER.

Finally, we tested uncharacterized proteins from the list. Among the six candidates, five had average TU:TO and TU:BL ratios > 2.0 (p<0.05). These yeast proteins were individually expressed in COS-7 cells and their localization compared to ER, mitochondria, and Golgi markers. YPR091C (recently annotated as Nvj2p) and YDL121C were predicted to be membrane proteins and appeared as an ER pattern with little co-localization with other markers. YMR209C contained a potential signal peptide and was found in the ER. YNL208W was predicted to be cytosolic but exhibited some ER association (*Figure 4—figure supplement 3*). In contrast, COS-7 cells failed to express YBL086C. These results generally validated the list and follow-up studies for these proteins are warranted.

## Discussion

Our quantitative proteomic analysis of the ER tubules identified a list of 79 candidates for ER tubule-enriched proteins. In addition to proteins known to shape the tubular network, including Rtn1p, Rtn2p, Yop1p, and Sey1p, we discovered proteins that function mainly in vesicle-based trafficking, membrane contact formation, lipid synthesis, or ER-related signaling. These findings emphasize the communication role of ER tubules, as they are the most outward reaching part of the organelle. They also support the notion that the ER tubules, lacking ribosomes on most of the surface, are broadly involved in lipid metabolism.

A unique feature of ER tubules compared to the rest of the ER is that it adopts high membrane curvature on its cross-section. Thus, curvature-sensitive functions or localization are expected for ER tubule-enriched proteins. For example, vesicle budding has been speculated to benefit from the membrane curvature observed in ER tubules. In yeast cells, most of the COPII vesicles, which transport cargo from the ER to Golgi, form in the tubular ER (*Okamoto et al., 2012*). In addition, proteins residing in ER tubules often possess membrane-anchoring motifs that prefer high curvature (*Zhang and Hu, 2016*). These motifs include TMH, such as in RTNs or REEP/Yop1p proteins, or an amphipathic helix (APH) that may insert easily into a curved membrane and is found in many of the proteins identified here.

### Organelle biogenesis

Our list contains a number of proteins participating in organelle biogenesis. Being the major source of intracellular lipids, the ER is expected to be involved in the formation of other membrane-bound compartments. For example, during de novo formation of peroxisomes, pre-peroxisomal vesicles (PPVs) are proposed to bud from the ER and mature to form functional peroxisomes (*Hoepfner et al., 2005*). We identified Pex30p (TU:TO 4.52; TU:BL 5.73) as an ER tubule-enriched protein. Pex30p has consistently been shown to contain a RHD, which is composed of two tandem TMHs; it is capable of tubulating membranes and may play a role in PPV biogenesis (*Joshi et al., 2016*).

Nuclear organization also depends on ER morphology. Tubule-shaping proteins play roles in the nuclear envelope (NE) reassembly after mitosis (*Dawson et al., 2009*). Furthermore, the membranes surrounding the nuclear pore have high curvature similar to that of ER tubules (*Mészáros et al., 2015*). POM33 (TU:TO 3.81; TU:BL 3.79) may provide additional linkage between nucleus biogenesis and ER tubules, as it has also been implicated in the nuclear pore complex distribution and NE remodeling (*Casey et al., 2015*; *Chadrin et al., 2010*; *Floch et al., 2015*). The localization of POM33 and its homologous proteins (PER33, Tts1, and TMEM33) can be explained by the presence of multiple TMHs and APHs and their interactions with known tubule proteins, including Rtn1p, Sey1p, and Lnp1p (*Casey et al., 2015*; *Chadrin et al., 2010*; *Zhang and Oliferenko, 2014*).

## Vesicular trafficking

The budding process of COPII vesicles may take advantage of the existing membrane curvature of ER tubules, and cargo sorting could utilize the vast reticular network as a platform. Our analysis failed to capture core components of the COPII coat. However, several known regulators of COPII formation were found. Components of the p24 family complex, Erp2p (TU:TO 3.12; TU:BL 2.88) and Erv25p (TU:TO 3.93; TU:BL 2.93), along with Erp1p (not included in our list, but with TU:TO 1.39, TU:BL 1.79, p<0.05) and Emp24p, sort GPI-anchored proteins into COPII vesicles (*Belden and Barlowe, 2001*; *D'Arcangelo et al., 2015*; *Schröder et al., 1995*). Emp47p (TU:TO 2.58; TU:BL 2.19), a yeast homolog of ERGIC-53, functions in glycoprotein secretion (*Sato and Nakano, 2002*). Pho86p (TU:TO 2.49; TU:BL 3.21) and Pho88p (TU:TO 5.68; TU:BL 3.24) are required for the export of phosphate transporter Pho84p (*James and Nachiappan, 2014*). These candidate proteins strongly support the notion that ER exit sites (ERESs), where ER cargo is sorted into COPII vesicles for exportation, are enriched in ER tubules.

## Organelle contact

ER tubules have been shown to make extensive contacts with other membranes, including the plasma membrane (PM), mitochondria, and endosomes (*Creutz et al., 2004*; *Du et al., 2011*; *Giordano et al., 2013*; *Kornmann et al., 2009*; *Toulmay and Prinz, 2012*; *West et al., 2011*). Such roles are expected as the tubular network is broadly distributed in the cell. Tricalbin proteins Tcb1p (TU:TO 2.49; TU:BL 1.57), Tcb2p (TU:TO 3.30; TU:BL 1.72), and Tcb3p (TU:TO 2.37; TU:BL 3.40) are homologs of extended synaptotagmins in mammals, which localize in the ER and associate with the PM (*Toulmay and Prinz, 2012*). Scs2p (TU:TO 4.01; TU:BL 3.19), a homolog of VAP-A, is also known for ER-PM contact (*Kim et al., 2015*). Another example of a contact protein residing in the ER tubule is YPR091C (TU:TO 4.07; TU:BL 3.37). It was renamed Nvj2p for its localization between the nucleus and vacuole (*Toulmay and Prinz, 2012*) and was recently found to form an inducible contact with Golgi (*Liu et al., 2017*). The existence of TMH in both Tcb proteins and Nvj2p is consistent with their commitment to ER tubules.

## Lipid synthesis

Lipid synthesis has long been thought to be the major role of smooth ER, which includes most of the tubular ER. We identified proteins that are related to several aspects of lipid synthesis, including enzymes along the ergosterol biogenesis pathway. Erg9p (TU:TO 4.60; TU:BL 2.57), or squalene synthase, acts in the earlier steps (*Jennings et al., 1991*), whereas Erg26p (TU:TO 3.61; TU:BL 3.34) and Erg27p (TU:TO 4.28; TU:BL 1.58) play roles in the later steps (*Gachotte et al., 1998*). Members of the conserved ER membrane protein complex (EMC) were also identified, including EMC1 (TU:TO 3.19; TU:BL 2.07), EMC2 (TU:TO 2.73; TU:BL 1.84), and EMC4 (TU:TO 2.34; TU:BL 1.58), the function of which are yet to be identified but likely related to lipid synthesis and/or transfer (*Lahiri et al., 2014*). Finally, transcriptional regulators of lipid metabolism were among the candidate proteins. Yet1p (TU:TO 2.44; TU:BL 2.67) and Yet3p (TU:TO 2.42; TU:BL 2.74) physically interact with Scs2p and the transcriptional repressor Opi1p (*Wilson et al., 2011*). The resulting complex plays a key regulatory role during inositol starvation. The ERG proteins contain predicted hydrophobic segments that may potentially behave as TMHs. Some of these proteins may also be targeted to the tubular ER by APHs. Alternatively, intermediates of lipid synthesis may favor the curved environment provided by the tubular ER network.

## Stress sensing

The stress sensing ability of the tubular ER is previously unidentified, and yet expected, as the network covers a substantial amount of space in the cell. Recently, ER stress was reported to be sensed and alleviated by Nvj2p (*Liu et al., 2017*), an ER tubule-enriched protein, as shown here. Additional stresses sensed by ER tubules that are inferred from our results include DNA integrity and chromatin structure (Msc1p, Bmh2p, Yim1p, Scp160p) and oxidation (Zta1p). ER tubules may also associate with acidic stress, as inferred from the quantification data of Yro2p (TU:TO 2.49, TU:BL 3.17), though the p-value for its TU:TO ratio (0.078) was beyond the cut-off (0.05). How the ER tubules contribute to these processes remains to be investigated.

## Application of the method

In this study, we successfully identified a list of ER tubule-enriched proteins. Nevertheless, certain candidates may have been missed due to a failure of ratio determination, that is, proteins are present in the TU but barely detected in the TO or BL. Additional candidates, especially proteins with low abundance, may be identified through extra replicates. Some ER tubule-based microsomes, likely tethered to other heavy membranes, may be lost during early purification steps. For example, the ER-mitochondria contact is mainly mediated by the ER mitochondrial encounter structure (ERMES) complex (*Kornmann et al., 2009*) and occurs predominantly in ER tubules (*Friedman and Voeltz, 2011*). Thus, the ER-localized ERMES component Mmm1p would be enriched in ER tubules, but it was not found in our list. To expand the inventory of ER tubule proteins, tubular marker-containing microsomes can be attempted to be isolated from total lysates.

The combination of immunoisolation and quantitative proteomics is efficient and useful for analyzing the components of ER tubules. Similarly, changes in the proteomic landscape due to specific treatment or genetic manipulation can easily be identified. The method can also be applied to proteomic studies of other organelles, suborganellar compartments, or other organisms. Furthermore, isolated membrane-bound structures can be used for equivalent high-throughput analysis, such as lipidomics.

# Materials and methods

## Constructs

For yeast cell expression, SEC63 and YOP1 were amplified with their own promoters and terminators, followed by different tags at their C-terminal, as indicated. These fragments were subcloned into pRS316 or pRS315. For mammalian cell expression, NSDHL, FDFT1, VAPA, DPM1, and EMC2 were amplified by PCR from the cDNA library for HeLa cells, and then ligated into PCI-neo-2HA vector. HT008 was amplified from the cDNA library for HeLa cells with a C-terminal HA tag and subcloned into pcDNA4/TO vector (Invitrogen). All constructs were confirmed by DNA sequencing.

## HAC1 transcript splicing analysis

Yeast cells were cultured at 30°C to an $OD_{600}$ of 1, and then treated with or without 4 µg/mL tunicamycin for 1 hr. Total RNA was extracted from 1.5 mL of cells using Yeast RNAiso Kit (TaKaRa). First-strand cDNAs were produced from yeast total RNA using M-MLV Reverse Transcriptase (Promega) and used as a template for amplification of HAC1 cDNA by PCR. ACT1 was used as an internal control. The PCR products were run on 1% (w/v) agarose gels for analysis.

## Yeast microsome preparation

Wild-type strain BY4741 (*MATa his3Δ1 leu2Δ met15Δ ura3Δ*) was transformed with the construct combination of pRS315:Sec63-HA and pRS313:Yop1-FLAG or pRS315:Sec63-GFP and pRS313:Yop1-STREP. The cells were cultured at 30°C in synthetic medium (-LEU and -HIS) to an $OD_{600}$ of approximately 1.

Approximately 10 mL of cultured cells were pelleted at 2300 x *g* (rotor FA-45-24-11, Eppendorf) at 4°C for 5 min, washed twice with 1.5 mL of 0.05 M EDTA (pH 8.0), resuspended in 1.0 mL of ETB buffer (0.05 M EDTA pH 8.0, 0.1 M Tris 9.0, 2.5% BME), and incubated at 30°C for 30 min with gentle shaking. After incubation, cells were pelleted and converted to spheroplasts by incubating with 1.5 mL of 2% snailase dissolved in sorbitol buffer (1 M sorbitol, 0.02 M sodium citrate, 0.1 M EDTA, 0.02 M $Na_2HPO_4$, pH 5.8) at 30°C for 1 hr. After 1 hr of enzymatic treatment, spheroplasts were washed twice with 1 mL of sorbitol buffer on ice. The washed spheroplasts were resuspended and swollen in 1.2 mL of lysis buffer 1 (800 mM sorbitol, 10 mM triethanolamine, 1 mM EDTA pH 8.0, and protease inhibitor cocktail) (*Klemm et al., 2009*) for 15 min on ice and homogenized with 35 strokes in a tight-fitted Dounce homogenizer. Crude homogenates (Total) were centrifuged at 1000 x *g* (rotor FA-45-24-11, Eppendorf) at 4°C for 5 min to remove the nucleus and unbroken cells. The supernatant (S1k) was further centrifuged at 20,000 x *g* (rotor FA-45-24-11, Eppendorf) at 4°C for 30 min to remove heavy organelle fractions. The resulting supernatant (S20k) was centrifuged at 100,000 x *g* (rotor TLA 100.3, Beckman) at 4°C for 40 min. The supernatant (S100k) was collected for Western blot analysis and the pellet (P100k) as the microsome fraction. For Western blot analysis,

the pellet (P100k) was boiled directly in 2x SDS-PAGE sample loading buffer. For mass spectrometric analysis, P100k was solubilized directly in 100 µL of 0.1 M ammonium bicarbonate (ABC) buffer containing 0.1% RapiGest SF and boiled at 95°C for 20 min.

## Mammalian microsome preparation

Cells were washed twice with ice-cold phosphate-buffered saline (PBS), collected in 1 mL of PBS by scraping, and pelleted by centrifugation at 6000 x $g$ (rotor FA-45-24-11, Eppendorf) at 4°C for 1 min. Cell pellets were resuspended and swollen in 1.2 mL of lysis buffer 2 (20 mM Hepes 7.4, 250 mM sucrose, 1 mM EDTA, and protease inhibitor cocktail) for 15 min on ice. The cells were then processed as described above (yeast microsome preparation).

## Immunoisolation of tubular ER

To isolate Yop1p-containing microsomes, P100k was resuspended in 700 µL of lysis buffer 1 plus 150 mM NaCl and placed on ice for 15 min. After incubation, 40 µL of anti-Flag affinity gel (Sigma, F2426) was added according to the manufacturer's instructions and the solution rotated for 2 hr at 4°C. Next, the affinity gel with bond vesicles was pelleted at 800 x $g$ (rotor FA-45-24-11, Eppendorf) at 4°C for 2 min and washed twice with 700 µL of lysis buffer 1 plus 150 mM NaCl. For Western blot analysis, the pellet was boiled directly in 2x SDS-PAGE sample loading buffer. For mass spectrometric analysis, the pellet was solubilized in 0.1 M ABC buffer containing 0.1% RapiGest SF and boiled for 20 min. The supernatant was collected for mass spectrometry. For the Percoll gradient assay, the pellet was resuspended in lysis buffer one containing Flag peptides (1 µg/µL) to elute the bonding vesicles.

To isolate REEP5-containing microsomes, P100k was resuspended in 700 µL of lysis buffer 2 plus 150 mM NaCl and placed on ice for 15 min. For HeLa cells (ATCC), rabbit anti-Reep5 antibody (1:500, Proteintech, 14643) was added and incubated for 1.5 hr at 4°C. Immobilized protein A/G (30 µL; Pierce, 53133) was then added and incubated for another 1.5 hr at 4°C. For Flp-In-293 cells (derived from ThermoFisher R78007), only 40 µL of anti-Flag affinity gel (Sigma, F2426) was used. Next, the gels were pelleted and washed twice with 700 µL of lysis buffer 2. The pellets were boiled directly in 2x SDS-PAGE sample loading buffer and analyzed by Western blot. To achieve a quantitative comparison, the Western blot bands were analyzed by ImageJ software.

## Percoll gradient assay

Thirty percent Percoll was prepared from a 100% Percoll solution by mixing 100% Percoll with 1xPBS at a ratio of 3:7 (volume). A total of 2 mL of the 30% Percoll solution was pipetted into a centrifugation tube. Total microsomes (P100k) resuspended in PBS or samples eluted from the immunoprecipitated beads (200 µL) were carefully layered on top of the Percoll solution. Centrifugation was performed at 95,000 x $g$ (rotor TLS55, Beckman) at 4°C for 40 min. After centrifugation, the gradient was fractionated into 21 tubes, 100 µL per fraction. The samples were then analyzed by Western blotting and quantified with ImageJ.

## Electron microscopy

Negative-stain EM was performed with 2% uranyl acetate solubilized in deionized water. TO samples were 100 µL of resuspended P100k (from 10 mL cultures) in lysis buffer 1. TU samples were immunoisolated microsomes (from 10 mL cultures) eluted by Flag peptide. The eluate was concentrated ~20 fold by Centrifugal Filters (Merck Millipore). A drop of 5 µL of sample solution was then placed onto a glow-discharged carbon-coated copper grid for 1 min. Excessive sample was removed by filter paper and the grids washed with one drop of deionized water and stained with one drop of fresh 2% uranyl acetate for 40 s. Images were collected at room temperature using a HITACHI HT7700 transmission electron microscope.

## Protein digestion and stable isotope dimethyl labeling

The concentration of total proteins eluted from the anti-Flag agarose gel or extracted from the microsome pellet was determined using the Bradford assay. The proteins (~10 µg) were reduced with 10 mM dithiothreitol (DTT) at 56°C for 40 min and alkylated with 20 mM iodoacetamide at room temperature in darkness for 40 min. An additional 10 mM DTT was added to consume the

excess iodoacetamide. Proteins were digested with trypsin (Promega, Madison, USA) at an enzyme-to-protein ratio of 1:100 (w/w) at 37°C for 3 hr, followed by the addition of fresh trypsin at 1:100 (w/w) and incubation at 37°C overnight. After quenching digestion by acidification (pH ≤2), the protein digest was centrifuged at 13,000 x *g* (rotor FA-45-24-11, Eppendorf) for 10 min. The supernatant was desalted with C18 microspin columns (The Nest Group, USA) and lyophilized under vacuum.

For triplex dimethyl labeling of peptides (*Boersema et al., 2009*), the tryptic digest of each sample (~10 μg) was suspended in 100 μL triethylammonium bicarbonate solution (TEAB, 100 mM, pH 8.0). Next, 4 μL of formaldehyde ($CH_2O$, $CD_2O$, and $^{13}CD_2O$, 4%, v/v) was added to specific samples for differential isotope labeling. The same volume of fresh 0.6 M cyanoborohydride ($NaBH_3CN$) was added to the light and intermediate labeled peptides, and the labeled $NaBD_3CN$ added to the heavy labeled peptides. After incubation for 1 hr at room temperature, 16 μL of 1% ammonium solution was added and vortexed for 10 min. Finally, 8 μL of 5% formic acid was added to quench the reaction. Each TU, TO, and BL sample was prepared in biological triplicates (n = 3). The TU, TO, and BL samples in each set of biological replicates were separately labeled with the light, intermediate, and heavy isotope mass labels. Each set of triplex labeled samples were then pooled at a 1:1:1 ratio. The pooled labeled peptides (~30 μg) were desalted and dried out by speed vacuum prior to nano-LC-MS/MS analysis.

## Nano-LC-MS/MS analysis

Peptide samples were analyzed by reverse-phase liquid chromatography electrospray ionization-MS/MS using an Eksigent Ultra Plus nano-LC system connected to a quadrupole time of flight Triple-TOF$^{TM}$ 5600 mass spectrometer (AB SCIEX). The vacuum-dried peptides were redissolved in solvent A (0.1% formic acid, 2% acetonitrile) and 1 μg of the peptide sample loaded onto the trapping column (10 mm ×100 μm, 5 μm C18 resin) at a flow rate of 5 μL/min for 20 min. Thereafter, the peptide mixtures were separated in the analytical C18-nano-capillary LC column (100 mm × 75 μm) packed in-house with C18-AQ 3 μm C18 resin (Dr. Maisch, GmbH, Germany). A 95 min gradient from 5% to 36% B (solvent B: 0.1% formic acid, 98% acetonitrile) at a flow rate of 300 nL/min was employed for peptide separation. The MS data acquisition was performed using Analyst TF 1.6 (AB SCIEX); the major source parameters were an ion spray voltage of 2400 V, 30 psi curtain gas, 10 psi ion source gas, and interface heater temperature of 150°C. The m/z range for MS and MS/MS scans was set from 350 to 1500 and 100 to 1500, respectively. In the IDA mode, the MS/MS spectra of the 40 most abundant parent ions were obtained following each MS1 survey scan with a 50 ms acquisition time per MS/MS scan. The mass width for dynamic exclusion was set to 50 mDa and the exclusion time 22 s.

## Proteomic data analysis

The data were processed by ProteinPilot Software v.4.5 (AB SCIEX) utilizing the Paragon and Pro-group Algorithm (*Shilov et al., 2007*). The software performs automatic recalibration such that typical mass errors for MS and MS/MS data were <10 ppm. The Uniprot *Saccharomyces cerevisiae* protein database (Jul-2013, 6629 entries) supplemented with the trypsin sequence and common protein contaminant sequences was employed. In the software algorithm, all modifications listed in UniMod are searched simultaneously with the tolerance specified as ±0.05 Da for peptides and MS/MS fragments (*Shilov et al., 2007*). ProteinPilot automatically clusters identified proteins into protein groups sharing common peptides. Only proteins identified with >99% confidence were retained, resulting in an FDR < 1% as calculated by a decoy database search. Dimethyl labeling was specified as the quantification method. Only peptides with unique sequences (not shared with other proteins) and free of miscleavage or variable modifications contributed to the protein ratio calculation. According to our isotopic labeling design, the relative TU:TO or TU:BL ratio for each identified protein was measured in triplicate and the average ratio and standard deviation calculated (full data in *Figure 3—source data 1*). Two-tailed t-tests were performed to determine p-values for the statistical deviation of individual protein ratios from unity. Proteins in pairwise comparisons of TU:TO and TU:BL were considered significantly up-regulated if the ratio was >2.0 and p<0.05. As for the TU:BL comparison, we relaxed the criteria to also include proteins with 1.4<TU:BL <2.0 and p<0.1.

## Immunofluorescence and confocal microscopy

COS-7 cells (ATCC) were cultured in Dulbecco's modified Eagle's medium supplemented with 10% fetal bovine serum at 37°C in 5% $CO_2$. For immunofluorescence experiments, cells were grown on 12-well plates for 24 hr and transfection performed using Lipofectamine 3000 (Invitrogen). Cells were fixed after 24 hr and permeabilized with 0.1% Triton X-100 (Bio Basic, Inc.). The cells were washed twice with PBS and blocked with 3% BSA for 1 hr at room temperature before probing with primary antibodies for 1 hr. The primary antibodies used were mouse anti-HA antibody (Abcam), rabbit anti-calreticulin antibody (Abcam), and mouse anti-climp63 antibody (EnzoLife Sciences). The cells were then incubated with various fluorophore-conjugated secondary antibodies for 1 hr. The secondary antibodies used were Alexa Fluor 488–conjugated anti–rabbit or Alexa Fluor 568–conjugated anti–mouse (Invitrogen). All images were captured at room temperature by a confocal microscope (Olympus FIUOVIEW FV100).

## Acknowledgements

We thank Yufeng Guo from Tianjin Institute of Industrial Biotechnology (CAS) for helping with data processing. WS is supported by grants from the Bairenjihua Program of the Chinese Academy of Sciences and the National Natural Science Foundation of China (No. 31401150). JH is supported by the National Key Research and Development Program (Grant No. 2016YFA0500201), the National Natural Science Foundation of China (Grants No. 31630020 and 3142100024), and an International Early Career Scientist grant from Howard Hughes Medical Institute.

## Additional information

### Funding

| Funder | Grant reference number | Author |
| --- | --- | --- |
| Howard Hughes Medical Institute | International Early Career Scientist grant | Junjie Hu |
| National Natural Science Foundation of China | 31630020 | Junjie Hu |
| Ministry of Science and Technology of the People's Republic of China | 2016YFA0500201 | Junjie Hu |
| National Natural Science Foundation of China | 31401150 | Wenqing Shui |
| Chinese Academy of Sciences | Bairenjihua Program | Wenqing Shui |
| National Natural Science Foundation of China | 3142100024 | Junjie Hu |

The funders had no role in study design, data collection and interpretation, or the decision to submit the work for publication.

### Author contributions

XW, Conceptualization, Data curation, Formal analysis, Validation, Visualization, Methodology; SL, Data curation, Formal analysis; HW, Formal analysis, Performed experiments in Figure 4-figure supplement 3; WS, Conceptualization, Data curation, Methodology, Writing—original draft, Writing—review and editing; JH, Conceptualization, Supervision, Funding acquisition, Validation, Writing—original draft, Project administration, Writing—review and editing

### Author ORCIDs

Junjie Hu, http://orcid.org/0000-0003-4712-2243

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
