## [Decision Letter]

Thank you for submitting your article "Quantitative proteomics reveal proteins enriched in tubular endoplasmic reticulum of *Saccharomyces cerevisiae*" for consideration by *eLife*. Your article has been reviewed by three peer reviewers, and the evaluation has been overseen by Randy Schekman as the Reviewing and Senior Editor. The following individual involved in review of your submission has agreed to reveal his identity: John J.M. Bergeron (Reviewer #2).

The reviewers have discussed the reviews with one another and the Reviewing Editor has drafted this decision to help you prepare a revised submission.

Summary:

In this Tools and Resources study, Wang et al. develop a new method to enrich ER microsomes and then immunoisolate tubular ER membranes in yeast using Flag-tagged Yop1 as the affinity tag. Quantitative proteomics is then used to identify other proteins enriched in this tubular ER prep. Starting with approximately 485 proteins that are readily detected in their microsome prep, 80 proteins were significantly enriched through Yop1 immunoisolation methods. The list of 80 proteins includes known tubular ER proteins (e.g. reticulons) and other candidates that may be involved in the structure/function of tubular ER. Overall this is a data-driven study that is straightforward and combines membrane isolation with quantitative proteomics. However, we have a number of specific concerns that you will need to satisfy if this work to be published in *eLife*. Regarding significance, it seems less clear how broadly applicable the method will be to other research labs. Cataloging the proteins that are enriched in tubular ER should be a useful resource.

Essential revisions:

1) A criticism of this reviewer is that all TU candidate proteins presented in Table 1 and two of the candidates highlighted in the Figure 4—figure supplement 3 were reported in a systematic localization study by Ghaemmighami et al. to localize to the entire ER including the nuclear envelope (NE) (see images at http://yeastgfp.yeastgenome.org). Indeed, one of the main characteristics of specific TU-ER proteins, such as the reference proteins Rtn1,2p, Sey1p, and Yop1p, is their exclusion from the flat membranes of the NE. The authors should visually verify the absence of their TU-ER candidates from the NE in yeast cells. If the proteins indeed localize to the NE as reported elsewhere, it would be difficult to understand how these candidates could have been found to be enriched in the TU-ER.

It is further rather unclear why Yop1p seems to be quantitatively absent from Sec63p marked ER-sheets. As also mentioned in the manuscript, several other studies have found the tubule shaping proteins on the edges of sheets. Thus, all Sec63p microsomes should contain at least some Yop1p that should likely suffice to pull them down by the antibody affinity technique.

In conclusion, the list of proteins might be of high interest as a resource for the field, however, its validity is as it stands is not entirely verified.

2) The coherence of Table 1 with [Supplementary-material SD1-data]-4 may also require some work; it is difficult to deduce this either from Table 1 or [Supplementary-material SD1-data]–[Supplementary-material SD4-data]. One issue that may be of relevance to the community using the methodology is the reliability of the reproducibility of the methodology. It may be safer for the authors to report on n=3 biological replicates for each set of data (mass spec and westerns). It appears that technical replicates have been mixed in with biological replicates. Having n=3 biological replicates for all data would assure the community of reproducibility. The technical replicate issue simply tests the authors' mass spec and Western blot platforms that are not expected to vary much when samples are run on them at the same time. However, the subtleties of homogenization and responses of different biological replicates of cells (yeast and mammalian) to protein expression will vary and why biological replicates may be useful for the community.

The authors may wish to consider whether in yeast and mammalian cells the expression of tagged proteins may have led to some protein misfolding. In Figure 4 and in supplements to this figure, tagged proteins are overexpressed in COS-7 cells to determine if candidate tubular ER proteins co-localize with calreticulin in mammalian cells. While these observations support their conclusions, there is no evidence provided that the tagged versions are functional and reflect localization of the endogenous proteins. Appropriate limitations to this analysis should be stated.

3) It would be helpful to provide more information on the methods used for yeast microsome preparation and tubular ER immunoisolation. This should include volume of cell cultures grown (or wet weight of cell pellet), volumes used for spheroplast and lysis treatments as well as volume used to resuspend the P100K. Some description of the volume of resuspended P100K and Anti-FLAG affinity gel used in immunoisolations would also be helpful.

4) Since this is designed to be a purification scheme, the yield and enrichment or depletion of some key marker proteins through the P100K and immunoisolation procedures in Figure 1 should be compiled. It seems surprising that the integral membrane proteins ALP and Tlg2 remain in the S100K and are not pelleted by this spin. Does the S20K fraction contain a high protein/membrane concentration? It is also not really clear why during a simple centrifugation at 100k x g Golgi membranes marked by the t-SNARE Tlg2p, and vacuoles marked by the Pho8 gene product ALP would remain quantitatively in the supernatant S100, while the low-density ER membranes sediment. How is this possible? Some mention of why these membrane proteins are not pelleted would be helpful.

The authors show by Western blot a similar distribution on Percoll gradients of Yop1p in microsomes and Yop1p vesicles. Is it reasonable to request the distribution of total protein across the Percoll gradients?

5) The supplementary tables in [Supplementary-material SD1-data]-4 may need some work. There are no detailed legends and the coherence of the 4 tables is not obvious nor their coherence with Table 1. Legends may be helpful for each of the tabs for each table and a detailed explanation for the headings for each column in each table. There may be some confusion from possible typos here. It is these tables that may be the value of the work so some attention here may be warranted. It was unclear as to where the data may be found for the tandem mass spec of the samples not isotope labeled with quantification by ion currents or spectral counts. The Methods section of the article implies this has been done.

The coherence of Table 1 with [Supplementary-material SD1-data]-4 may also require some work; it is difficult to deduce this either from Table 1 or [Supplementary-material SD1-data]–[Supplementary-material SD4-data].

Other comments:

1) In general, the main text on the description of the proteomics should be simplified by using clearer language, spell out abbreviations, and avoid the use of terms that are usually not used in the field. E.g. what does the term "significant abundance ratio" mean (subsection “Proteomic analysis of ER tubule-enriched protein constituents”, last paragraph)? Also, the main selection criterion for TU proteins described and the following explanation of what the authors call "demonstration of sufficient quantification" is unfortunately even for experts in the field rather inscrutable. Please change the text.

2) One issue that may be of relevance to the community using the methodology is the reliability of the reproducibility of the methodology. It may be safer for the authors to report on n=3 biological replicates for each set of data (mass spec and westerns). It appears that technical replicates have been mixed in with biological replicates. Having n=3 biological replicates for all data would assure the community of reproducibility. The technical replicate issue simply tests the authors' mass spec and Western blot platforms that are not expected to vary much when samples are run on them at the same time. However, the subtleties of homogenization and responses of different biological replicates of cells (yeast and mammalian) to protein expression will vary and why biological replicates may be useful for the community.

3) The authors have no doubt considered EM of their isolated Yop1p structures – it would be helpful to have these with some quantification. Finally, the authors may consider a reassessment of the immunofluorescence of COS7 cells as indicated in greater detail below. Is there a reason that HeLa cells have not been done as well?

4) The authors show by Western blot a similar distribution on Percoll gradients of Yop1p in microsomes and Yop1p vesicles. Is it reasonable to request the distribution of total protein across the Percoll gradients?

5) By Western blot, the authors show a marked diminishment of Kar2p in microsomes with high levels in the final supernatant. The authors ascribe this to leakage but Kar2p secretion may also be a stress response. Have other stress markers been assessed? This might be expected if a proportion of the expressed proteins are misfolded. Yop1p also shows diminishment in microsomes and appreciable abundance in the final supernatant. This is contrary to the text that indicates that it was mostly in the P100K fraction. The low abundance in the P100K fraction, abundance in the final supernatant would be consistent with misfolded Yop1P. Figure 1—figure supplement 1 has not been described in the text as to exactly what it shows.

6) In Figure 1, the authors show the requirement of anti-FLAG to isolate the Yop1p vesicles tagged with FLAG. They also show very little Sey1p in the anti-FLAG immunoisolates but appreciable Dpm1p. There appears to be a marked discrepancy in the amount of Yop1p in the unbound fraction of Figure 1 compared to Figure 1. Is there an explanation? The text in the results describing detergent effects for Figure 1 may be confusing although the effect of rapigest is indicated to elute Yop1p in Figure 1—figure supplement 1. This section was confusing.

7) The summary of the isotope labelling experiments indicated in Figure 3 show differences in the enrichment of proteins in the Yop1P immunoisolated vesicles as compared to microsomes. Is there any enrichment over the total homogenates since neither of the proteins are enriched in microsomes to begin with? Where would dpm1p be in Figure 3? [Supplementary-material SD1-data] that summarizes the data is without a legend or description of what is in each column for each of the 4 tabs and how the data have been ordered. The finding of Pex30 may actually be an important discovery but its significance is impossible to deduce from the table as submitted. What is the significance of the meiotic protein, ribosomal protein S17 etc.? Are the proteins in [Supplementary-material SD4-data] all low abundance contaminants from other structures in the cell?

8) For the localization studies in COS7 cells the authors indicate that dpm1 and emc2 are soluble proteins- they are both integral membrane proteins. Coherence between Table 1 and [Supplementary-material SD1-data]-4 may not be obvious. Both the immunofluorescence and localizations in COS7 cells are difficult to interpret without a positive control for a tubular ER marker (CLIMP63 may link microtubules to ER) or an explanation for what may be incorrect localizations of possibly misfolded tagged dpm1 and tagged emc2.

---

## [Author Response]

Essential revisions:

*1) A criticism of this reviewer is that all TU candidate proteins presented in Table 1 and two of the candidates highlighted in the Figure 4—figure supplement 3 were reported in a systematic localization study by Ghaemmighami et al. to localize to the entire ER including the nuclear envelope (NE) (see images at http://yeastgfp.yeastgenome.org). Indeed, one of the main characteristics of specific TU-ER proteins, such as the reference proteins Rtn1,2p, Sey1p, and Yop1p, is their exclusion from the flat membranes of the NE. The authors should visually verify the absence of their TU-ER candidates from the NE in yeast cells. If the proteins indeed localize to the NE as reported elsewhere, it would be difficult to understand how these candidates could have been found to be enriched in the TU-ER.*

We think that not all proteins that are enriched in the tubular ER would necessarily be excluded from the NE. Though some proteins, such as Yop1p and Rtn1p, have strong preference for curved membranes and are less likely to reside in the NE, other proteins may interact with TU-exclusive proteins and be enriched in the tubular ER. In the latter case, these proteins may very likely enter the NE when they are not bound to TU-exclusive proteins. For example, the mammalian homolog of Pom33p (identified in our list, interacts with Rtn1p) clearly localizes to both the tubular ER and the NE (PubMed IDs: 25612671 and 26268696), and Sey1p has been found in the NE to regulate nuclear pore complex integrity (PubMed ID: 26041935). Furthermore, low levels of TU-exclusive proteins can be seen in the NE of yeast cells (16469703), possibly existing in highly curved membranes around nuclear pores.

It is further rather unclear why Yop1p seems to be quantitatively absent from Sec63p marked ER-sheets. As also mentioned in the manuscript, several other studies have found the tubule shaping proteins on the edges of sheets. Thus, all Sec63p microsomes should contain at least some Yop1p that should likely suffice to pull them down by the antibody affinity technique.

We agree with the reviewer that some Yop1p could be isolated with Sec63p because Yop1p marks the edges of ER sheets and/or Sec63p partly localizes in the tubular ER. However, Yop1p in sheets would represent a minor proportion of the total Yop1p and likely be present in very low abundance compared to other proteins in Sec63p-enriched microsomes. The same applies to Sec63p in tubules. In addition, the edges of sheets could rupture into relatively small microsomes that contain less Sec63p than regular sheet-derived microsomes. This would explain why we barely see Yop1p in Sec63p-tagged microsomes.

*In conclusion, the list of proteins might be of high interest as a resource for the field, however, its validity is as it stands is not entirely verified.*

*2) The coherence of Table 1 with [Supplementary-material SD1-data]-4 may also require some work; it is difficult to deduce this either from Table 1 or [Supplementary-material SD1-data]–[Supplementary-material SD4-data]. One issue that may be of relevance to the community using the methodology is the reliability of the reproducibility of the methodology. It may be safer for the authors to report on n=3 biological replicates for each set of data (mass spec and westerns). It appears that technical replicates have been mixed in with biological replicates. Having n=3 biological replicates for all data would assure the community of reproducibility. The technical replicate issue simply tests the authors' mass spec and Western blot platforms that are not expected to vary much when samples are run on them at the same time. However, the subtleties of homogenization and responses of different biological replicates of cells (yeast and mammalian) to protein expression will vary and why biological replicates may be useful for the community.*

We thank the reviewer for this suggestion. We have reformatted the supplementary tables to include the commonly known names of the proteins, which are now in line with Table 1.

For quantitative mass spec data, we have retained three biological replicates as suggested. The results shown in Figure 3 and the supplementary tables have been adjusted accordingly. One protein (nuclear protein Sth1p) has been deleted from the final list.

The authors may wish to consider whether in yeast and mammalian cells the expression of tagged proteins may have led to some protein misfolding. In Figure 4 and in supplements to this figure, tagged proteins are overexpressed in COS-7 cells to determine if candidate tubular ER proteins co-localize with calreticulin in mammalian cells. While these observations support their conclusions, there is no evidence provided that the tagged versions are functional and reflect localization of the endogenous proteins. Appropriate limitations to this analysis should be stated.

We agree with the reviewer that some mis-folding may occur. We have now tested the endogenous levels of some candidates. We generated an Flp-in cell line that stably expresses low levels of Flag-tagged REEP5, the mammalian homolog of Yop1p, and performed the same cell fractionation and immunoisolation experiments. Endogenous HT008, NSDHL, and VAPA are relatively enriched in TU microsomes. These results have been added as Figure 4—figure supplement 3.

*3) It would be helpful to provide more information on the methods used for yeast microsome preparation and tubular ER immunoisolation. This should include volume of cell cultures grown (or wet weight of cell pellet), volumes used for spheroplast and lysis treatments as well as volume used to resuspend the P100K. Some description of the volume of resuspended P100K and Anti-FLAG affinity gel used in immunoisolations would also be helpful.*

We thank the reviewer for this suggestion. We have modified the Methods accordingly (subsection “Yeast microsome preparation”).

*4) Since this is designed to be a purification scheme, the yield and enrichment or depletion of some key marker proteins through the P100K and immunoisolation procedures in Figure 1 should be compiled. It seems surprising that the integral membrane proteins ALP and Tlg2 remain in the S100K and are not pelleted by this spin. Does the S20K fraction contain a high protein/membrane concentration? It is also not really clear why during a simple centrifugation at 100k x g Golgi membranes marked by the t-SNARE Tlg2p, and vacuoles marked by the Pho8 gene product ALP would remain quantitatively in the supernatant S100, while the low-density ER membranes sediment. How is this possible? Some mention of why these membrane proteins are not pelleted would be helpful.*

The results for ALP and Tlg2 are surprising but reproducible. One possibility is that most of the vacuoles or Golgi were ruptured into small vesicles. Alternatively, the proteins may be cleaved off or detached from membranes during processing. We have mentioned these possibilities in the text.

*The authors show by Western blot a similar distribution on Percoll gradients of Yop1p in microsomes and Yop1p vesicles. Is it reasonable to request the distribution of total protein across the Percoll gradients?*

We added a silver stain gel showing the distribution of total protein across the Percoll gradients in Figure 2—figure supplement 1.

*5) The supplementary tables in [Supplementary-material SD1-data]-4 may need some work. There are no detailed legends and the coherence of the 4 tables is not obvious nor their coherence with Table 1. Legends may be helpful for each of the tabs for each table and a detailed explanation for the headings for each column in each table. There may be some confusion from possible typos here. It is these tables that may be the value of the work so some attention here may be warranted. It was unclear as to where the data may be found for the tandem mass spec of the samples not isotope labeled with quantification by ion currents or spectral counts. The Methods section of the article implies this has been done.*

We thank the reviewer for this suggestion. We have reformatted the supplementary tables accordingly. In this study, we only employed dimethyl isotope labeling for protein identification and quantification; no ion currents or spectral count-based methods were used for label-free quantification.

*The coherence of Table 1 with Figure 2—source data 1-4 may also require some work; it is difficult to deduce this either from Table 1 or Figure 2—source data 1-4.*

We have renamed one protein in Table 1 according to the standard name for SGD. We have also added standard protein names in [Supplementary-material SD3-data].

*Other comments:*

*1) In general, the main text on the description of the proteomics should be simplified by using clearer language, spell out abbreviations, and avoid the use of terms that are usually not used in the field. E.g. what does the term "significant abundance ratio" mean (subsection “Proteomic analysis of ER tubule-enriched protein constituents”, last paragraph)? Also, the main selection criterion for TU proteins described and the following explanation of what the authors call "demonstration of sufficient quantification" is unfortunately even for experts in the field rather inscrutable. Please change the text.*

We have rewritten these parts.

*2) One issue that may be of relevance to the community using the methodology is the reliability of the reproducibility of the methodology. It may be safer for the authors to report on n=3 biological replicates for each set of data (mass spec and westerns). It appears that technical replicates have been mixed in with biological replicates. Having n=3 biological replicates for all data would assure the community of reproducibility. The technical replicate issue simply tests the authors' mass spec and Western blot platforms that are not expected to vary much when samples are run on them at the same time. However, the subtleties of homogenization and responses of different biological replicates of cells (yeast and mammalian) to protein expression will vary and why biological replicates may be useful for the community.*

We have changed the data sets to include just three biological repeats (see above).

*3) The authors have no doubt considered EM of their isolated Yop1p structures – it would be helpful to have these with some quantification. Finally, the authors may consider a reassessment of the immunofluorescence of COS7 cells as indicated in greater detail below. Is there a reason that HeLa cells have not been done as well?*

We have performed negative-stain EM on “TO” and “TU” samples. As expected, “TU” microsomes are relatively small in size compared to “TO” microsomes. Interestingly, some short tubules were observed with “TU” samples, confirming the enrichment of tubule-derived microsomes. The diameters (~30 nm) of these microsomes are consistent with those of ER tubules seen in yeast cells. These data were added as Figure 2.

We chose COS-7 cells instead of HeLa cells because the tubular ER network is more prominent in COS-7 cells.

*4) The authors show by Western blot a similar distribution on Percoll gradients of Yop1p in microsomes and Yop1p vesicles. Is it reasonable to request the distribution of total protein across the Percoll gradients?*

See our answer to #4 above.

5) By Western blot, the authors show a marked diminishment of Kar2p in microsomes with high levels in the final supernatant. The authors ascribe this to leakage but Kar2p secretion may also be a stress response. Have other stress markers been assessed? This might be expected if a proportion of the expressed proteins are misfolded. Yop1p also shows diminishment in microsomes and appreciable abundance in the final supernatant. This is contrary to the text that indicates that it was mostly in the P100K fraction. The low abundance in the P100K fraction, abundance in the final supernatant would be consistent with misfolded Yop1P. Figure 1—figure supplement 1 has not been described in the text as to exactly what it shows.

We have tested ER stress in cells expressing Yop1p and Sec63p. Only a very minor splicing of HAC1 mRNA was observed, a general indication of ER stress in yeast cells. Consistently, Kar2p is not upregulated in these cells. Therefore, even if misfolded proteins accumulate, it is likely a very small proportion. These new supporting data were added as Figure 1—figure supplement 1.

We apologize for the confusing anti-Flag blots in Figure 1. The experiment was repeated using frozen samples; the updated results show that the majority of Yop1p ends up in P100K. The current blot would be most representative among numerous repeats of the same experiment. Similar patterns can be seen in Figure 4—figure supplement 3 for REEP5 tested in mammalian cells.

Figure 1—figure supplement 1 shows the expression of Flag-tagged Yop1p and HA-tagged Sec63p from three different clones, and we have referenced this in the text.

*6) In Figure 1, the authors show the requirement of anti-FLAG to isolate the Yop1p vesicles tagged with FLAG. They also show very little Sey1p in the anti-FLAG immunoisolates but appreciable Dpm1p. There appears to be a marked discrepancy in the amount of Yop1p in the unbound fraction of Figure 1 compared to Figure 1. Is there an explanation? The text in the results describing detergent effects for Figure 1 may be confusing although the effect of rapigest is indicated to elute Yop1p in Figure 1—figure supplement 1. This section was confusing.*

The original panels on the right side of Figure 1 were performed using relatively old anti-Flag affinity gel. To improve the efficiency of the immunoisolation and ensure consistency between figures, we have repeated the experiments with fresh anti-Flag beads. As shown in the new Figure 1, the efficiency of Flag precipitation and Sey1p co-precipitation was improved. Furthermore, Sey1p appears to be less enriched in the tubular ER than other proteins identified here (Figure 3). One possibility is that Sey1p tends to form puncta at three-way junctions, and junction-derived microsomes may not be equally represented in our samples.

When discussing detergents and IP efficiency, we meant detergents commonly used in IP experiments, such as Triton X-100; these relatively mild detergents may help expose epitope tags during IP. In contrast, RapiGest is a mass spec-compatible detergent that dissolves everything after immunoisolation. We have clarified this in the text.

*7) The summary of the isotope labelling experiments indicated in Figure 3 show differences in the enrichment of proteins in the Yop1P immunoisolated vesicles as compared to microsomes. Is there any enrichment over the total homogenates since neither of the proteins are enriched in microsomes to begin with? Where would dpm1p be in Figure 3? [Supplementary-material SD1-data] that summarizes the data is without a legend or description of what is in each column for each of the 4 tabs and how the data have been ordered. The finding of Pex30 may actually be an important discovery but its significance is impossible to deduce from the table as submitted. What is the significance of the meiotic protein, ribosomal protein S17 etc.? Are the proteins in [Supplementary-material SD4-data] all low abundance contaminants from other structures in the cell?*

In solution, there is no enrichment of TU proteins between total homogenates and total microsomes. However, when performing quantitative mass spec, samples are mixed with equal amounts of total proteins. Many proteins have been separated from microsomal proteins during stepped spins. Therefore, we assume that the TU proteins identified here would be further enriched compared to total homogenates. In other words, for each TU protein, the TU:TO^homogenates^ ratio is expected to be higher than the TU:TO^microsomes^ ratio. Dmp1 (TU:TO 4.06; TU:BL 4.27) would be ranked in the top 20-25% in the list.

[Supplementary-material SD1-data] has been reformatted to include clear labels. We have also sorted [Supplementary-material SD3-data] based on the TU:BL ratio to demonstrate specificity ranking. If sorted by TU:TO ratio, which shows relative enrichment in the tubular ER, a similar pattern is seen. Pex30 (TU:TO 4.07; TU:BL 3.37) is ranked similar to that of Dpm1 (see above).

Meiotic sister chromatid recombination protein 1 (MSC1) has been confirmed by high throughput localization studies as an ER-resident protein, but its specific function is unclear. One possible explanation for its enrichment in tubular ER is that, during interphase, when its activity is not needed, it may be kept in ER tubules where access to the nucleus is less likely. We do not have an explanation for ribosomal protein RL17b, as it is likely a false positive.

[Supplementary-material SD4-data] lists proteins that have a positive TU:TO ratio but are not found in BL. These proteins can be contaminants with low abundance or very specific candidates. The latter is the reason why we show these proteins as a reference.

*8) For the localization studies in COS7 cells the authors indicate that dpm1 and emc2 are soluble proteins- they are both integral membrane proteins. Coherence between Table 1 and [Supplementary-material SD1-data]-4 may not be obvious. Both the immunofluorescence and localizations in COS7 cells are difficult to interpret without a positive control for a tubular ER marker (CLIMP63 may link microtubules to ER) or an explanation for what may be incorrect localizations of possibly misfolded tagged dpm1 and tagged emc2.*

We have added standard protein names (based on SGD) in [Supplementary-material SD3-data] to match those in Table 1. Unlike yeast Dpm1p, mammalian DPM1 was predicted to contain no transmembrane region (PubMed ID: 9223280). Similarly, mammalian EMC2 (also called TTC35) exhibited a cytosolic distribution (PubMed ID: 22119785; Figure 2). We have also added zoomed in views for all immunofluorescence tests performed in COS-7 cells.